# Can Large Reasoning Models do Analogical Reasoning under Perceptual Uncertainty?

**Giacomo Camposampiero**\*                                    GIACOMO.CAMPOSAMPIERO1@IBM.COM
*IBM Research - Zurich and ETH Zürich*

**Michael Hersche**\*                                    MICHAEL.HERSCHE@IBM.COM
*IBM Research - Zurich*

**Roger Wattenhofer**                                    WATTENHOFER@ETHZ.CH
*ETH Zürich*

**Abu Sebastian**                                    ASE@ZURICH.IBM.COM
*IBM Research - Zurich*

**Abbas Rahimi**                                    ABR@ZURICH.IBM.COM
*IBM Research - Zurich*

**Editors:** Leilani H. Gilpin, Eleonora Giunchiglia, Pascal Hitzler, and Emile van Krieken

## Abstract

This work presents a first evaluation of two state-of-the-art Large Reasoning Models (LRMs), OpenAI's o3-mini and DeepSeek R1, on analogical reasoning, focusing on well-established nonverbal human IQ tests based on Raven's progressive matrices. We benchmark with the I-RAVEN dataset and its extension, I-RAVEN-X, which tests the ability to generalize to longer reasoning rules and ranges of the attribute values. To assess the influence of visual uncertainties on these symbolic analogical reasoning tests, we extend the I-RAVEN-X dataset, which otherwise assumes an oracle perception. We adopt a two-fold strategy to simulate this imperfect visual perception: 1) we introduce confounding attributes which, being sampled at random, do not contribute to the prediction of the correct answer of the puzzles, and 2) we smoothen the distributions of the input attributes' values. We observe a sharp decline in OpenAI's o3-mini task accuracy, dropping from 86.6% on the original I-RAVEN to just 17.0%—approaching random chance—on the more challenging I-RAVEN-X, which increases input length and range and emulates perceptual uncertainty. This drop occurred despite spending $3.4\times$ more reasoning tokens. A similar trend is also observed for DeepSeek R1: from 80.6% to 23.2%. On the other hand, a neuro-symbolic probabilistic abductive model, ARLC, that achieves state-of-the-art performances on I-RAVEN, can robustly reason under all these out-of-distribution tests, maintaining strong accuracy with only a modest accuracy reduction from 98.6% to 88.0%. Our code is available at https://github.com/IBM/raven-large-language-models.

## 1. Introduction

Large Language Models (LLMs) such as GPT-4 (OpenAI et al., 2024), Gemini (Gemini Team et al., 2024), and Claude (Anthropic, 2024) have demonstrated great proficiency in generating fluent and contextually relevant text. However, their capabilities in more complex domains, such as reasoning and planning, have been shown to be brittle even in simple

---

\* Equal contribution

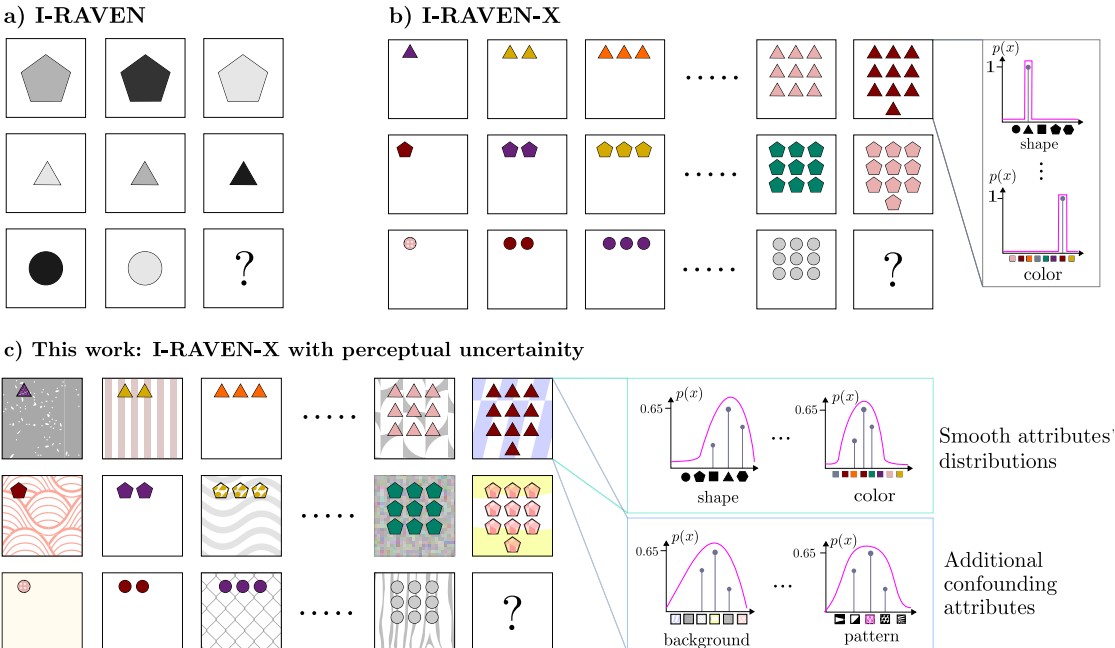

Figure 1: **Analogical reasoning under perceptual uncertainty.** This figure highlights all the different axes of generalization and robustness to uncertainty, which I-RAVEN-X stresses. Compared to standard I-RAVEN **(a)**, I-RAVEN-X **(b)** involves more panels per row (10 vs. 3) and larger attribute dynamic ranges (up to $100\times$ more values per attribute). This work **(c)** introduces uncertainty in the reasoning process through confounders (such as panels' background and color patterns within objects) and smoothening the attribute values' distributions (displayed on the right for the panel in position $(1, 10)$). We adopt a visual representation of the panels and their attributes for clarity of explanation; in practice, however, our dataset is purely symbolic and has not been extended yet to the visual domain.

tasks (Gendron et al., 2024; Wu et al., 2024), fail to attain levels of general abstract reasoning comparable to humans (Odouard and Mitchell, 2022; Thomm et al., 2024; Lewis and Mitchell, 2025; Camposampiero et al., 2023) and may be in some cases a result of data contamination (Roberts et al., 2023; Mirzadeh et al., 2025). To mitigate this issue, the attention has shifted from training-time to test-time compute scaling. This resulted in the development of a new generation of systems, dubbed Large Reasoning Models (LRMs), that can dynamically allocate variable compute at test time based on the input query. Unlike LLMs, which behaved mostly like approximate retrievers, LRMs such as OpenAI's o1 and o3 (OpenAI, 2024), DeepSeek R1 (DeepSeek-AI et al., 2025), and Qwen QwQ (Qwen Team, 2024) approach reasoning tasks by exploring the space of solutions through pseudo-actions using Chain-of-Thought (CoT) (Wei et al., 2022) tokens. Contrary to LLMs (Webb et al., 2023; Hu et al., 2023; Hersche et al., 2025; Moskvichev et al., 2023; Mitchell et al., 2024; Lewis and Mitchell, 2025), however, the analogical reasoning abilities of LRMs have not yet been extensively evaluated ($L_1$), with only Latif et al. (2024) showing limited results

on it. Furthermore, we identify a critical limitation of recent works that benchmarked language models on *visual* analogical reasoning. Due to their poor performance in the visual domain (Mitchell et al., 2024; Jiang et al., 2024b; Cao et al., 2024; Ahrabian et al., 2024b; Zhang et al., 2024b), assuming the availability of an *oracle perception* and prompt LLMs with discrete symbolic transcriptions of the test examples has become a standard practice (Webb et al., 2023; Hu et al., 2023; Hersche et al., 2025). However, this assumption bypasses crucial steps in visual analogical reasoning. First, the oracle perception takes the knowledge about the set of attributes needed for the reasoning task for granted, avoiding the step of filtering attributes that are irrelevant to the answer prediction ($L_2$). Second, the oracle perception usually provides the attribute values with full confidence, an unrealistic assumption since any perception model will always feature some degree of uncertainty ($L_3$). In order to tackle these issues, this paper makes the following contributions:

1. In Section 3, we introduce a more thorough method to benchmark analogical reasoning in LRMs, extending the fully-symbolic dataset I-RAVEN-X (Hersche et al., 2025) with simulated perceptual uncertainty given by confounding attributes (tackling $L_2$) and uncertainty on the reasoning attributes' values (tackling $L_3$), as shown in Figure 1. The resulting dataset, I-RAVEN-X, improves on the standard I-RAVEN (Hu et al., 2021) by allowing to test a) *productivity* (longer reasoning relations), *systematicity* (more values for each attribute), *robustness to confounding factors* (the set of generative attributes in RPM is augmented with randomly sampled values, which do not contribute to the reasoning), and *robustness to non-degenerate value distributions* (uncertainty associated to the value of each attribute).

2. In Section 5, we present the first exhaustive analysis on two state-of-the-art (SOTA) LRMs, OpenAI's o3-mini and DeepSeek R1, on symbolic analogical reasoning using the newly-introduced I-RAVEN-X (addressing $L_1$). Despite showing significant improvement over LLMs, LRMs are affected by a a sharp decay in accuracy when simultaneously evaluated on all generalization dimensions (productivity, systematicity, and robustness), with o3-mini and R1 dropping from 86.6% to 17.0% and 80.6% to 23.2%, respectively, substantially approaching random chance.

3. In Section 4.2, we propose a novel entropy-based regularizer integrable in abductive reasoning frameworks to improve their robustness to noisy features, implicitly allowing them to filter confounders in the decision process. We implement this regularizer in ARLC (Camposampiero et al., 2024) and show that it helps retaining high accuracy in harsh signal-to-noise ratio (SNR) conditions (up to $-20$ dB). Furthermore, we show that ARLC, as a neuro-symbolic probabilistic abductive reasoner supporting reasoning under uncertainty, achieves significantly stronger accuracy (88.3%) compared to LRMs.

## 2. Related works

**Analogical reasoning benchmarks.** A wide range of benchmarks to assess abstract reasoning has been proposed in the past decade (Bilker et al., 2012; Cherian et al., 2023; Chollet, 2019; Niedermayr et al., 2024). Raven's Progressive Matrices (RPM) (Raven et al., 1938; Carpenter et al., 1990; Bilker et al., 2012) is one of the most prominent among them

due to its extensive use to benchmark for abstract reasoning, analogy-making, and out-of-distribution (OOD) testing (Benny et al., 2021; Hu et al., 2021; Małkiński and Mańdziuk, 2025; Mitchell, 2021; Zhang et al., 2019). RAVEN (Zhang et al., 2019) represented one of the first attempts to build a dataset in the context of RPM that aimed at associating vision with structural, relational, and analogical reasoning in a hierarchical representation. I-RAVEN (Hu et al., 2021) (Figure 1a) improved RAVEN, proposing a new generation algorithm based on attribute bisection trees. This ensured that candidate panels are sampled from an unbiased candidate set, avoiding shortcut solutions that were possible in the original dataset. I-RAVEN-X (Hersche et al., 2025) extended I-RAVEN, introducing a parameterizable number of columns and a dynamic range of attributes that allow testing the generalization of analogical reasoning to longer reasoning chains and an increased number of concepts. In Figure 1b, this can be identified by the larger number of columns and a wider range of attributes (such as the color and the number of objects), respectively. In addition, the dataset was narrowed down to a single constellation (`center`, containing one object per panel) which was observed to be simultaneously a strong test for a wide range of logical and arithmetic skills and unexpectedly challenging for LLMs (Hersche et al., 2025).

**Large reasoning models.** Recent research has focused on training LLMs to exhibit human-like reasoning (OpenAI, 2024), yet a major obstacle remains the scarcity of annotated, step-by-step reasoning data. To overcome this issue, researchers started transitioning to LLM-driven search algorithms that automatically generate accurate reasoning trajectories through external verification (Luo et al., 2024) and RL-based techniques (Zhang et al., 2024a; Shao et al., 2024). Moreover, scaling test-time computation was also shown to be useful to refine intermediate reasoning steps and further improve accuracy on reasoning tasks (Snell et al., 2024). Together, the combination of RL-driven train-time scaling and search-based test-time scaling paved the way for a new generation of systems, Large Reasoning Models (LRMs), with significantly enhanced reasoning performance (Xu et al., 2025).

**Neuro-symbolic architectures for RPM.** Improving on monolithic deep learning models (Wu et al., 2020; Benny et al., 2021), neuro-symbolic architectures implementing abductive reasoning (Magnani, 2009) achieved remarkable success, scoring SOTA results on this analogical reasoning test. Initially introduced with the PrAE learner (Zhang et al., 2021), this approach was further improved by the NVSA model (Hersche et al., 2023). In NVSA, the probabilistic reasoning was implemented via distributed representations and operators of vector-symbolic architectures (VSAs) (Gayler, 2003; Kanerva, 2009; Plate, 1995). VSAs, besides their computational and scalability benefits, provide a common language with neural networks for better interface and deeper integration. Both PrAE and NVSA are examples of `Neuro | Symbolic` networks (Kautz, 2022); that is, they are systems composed of a neural vision module that interacts with a static symbolic reasoning system through a well-defined interface. Follow-up works extended these systems, mostly moving from pure knowledge representations to more trainable architectures that can learn from examples to reason and improve their expressiveness (Zhang et al., 2022; Camposampiero et al., 2024; Sun et al., 2025). Some of them, as ARLC (Camposampiero et al., 2024), could be classified as `Neuro[Symbolic]` systems (Kautz, 2022) since the reasoning rules are learned in a differentiable fashion from a generic rule template encoded in distributed representations, and it is capable of combinatorial reasoning by exploiting computation-in-superposition.

## 3. Integrating perceptual uncertainty into I-RAVEN-X

Contrary to the standard I-RAVEN, I-RAVEN-X is a fully symbolic benchmark that evaluates analogical reasoning under the assumption of an oracle perception. However, this is a rather strong assumption that neglects the uncertainty that would necessarily result from the extraction of those attributes in real-world scenarios and their influence on the analogical reasoning process. In this work, we propose an extension of I-RAVEN-X to overcome it, augmenting the original dataset by:

1. integrating *confounding attributes* for each RPM example, and

2. *smoothening* the original degenerate attribute values' distributions.

Together, 1. and 2. allow us to loosen the strong assumption of an oracle perception, simulating an imperfect perception front-end while retaining the main advantage of operating in a purely symbolic setting (that is, leveraging text-based models rather than their weaker multi-modal equivalents).

### 3.1. Confounding attributes

Confounding attributes represent properties and patterns that can be extracted from the visual inputs by a front-end perception module but are not relevant to the reasoning process. This could be the case, for instance, when the attributes are extracted by unsupervised vision models such as Variational Autoencoders (Kingma and Welling, 2013) or even a multi-modal LLM that is prompted to extract the attributes. In Figure 1c, confounding attributes are represented by the background of the input panels and the color patterns, which sometimes appear inside the objects. While the original RAVEN dataset includes *noise attributes* (e.g., the orientation), we argue that these are not real confounders as they do not add any noise to the RPM tests (refer to Appendix C for a more extensive discussion). In I-RAVEN-X, we integrate confounders by extending the set of original attributes of each panel with an arbitrary number of additional attributes uniformly sampled in the interval $[0, m-1]$, where $m$ is the range of the attributes' values. For large enough $m$, the probability of sampling values that fit a valid rule is negligible, and hence, confounders do not introduce ambiguities in the choice of the answer panel. However, they linearly reduce the SNR in the reasoning process, requiring models to implement strategies to filter out noisy input components.

### 3.2. Smooth attribute values' distributions

We deviate from the original I-RAVEN-X degenerate attributes' distributions and introduce variance, which allows us to test the robustness of the models when reasoning with uncertain attributes' values. Figure 1 highlights this relaxation, from one-hot probability mass functions (PMFs) of the standard I-RAVEN-X (Figure 1b) to the distributed PMFs of our proposed extension (Figure 1c). In practice, we smoothen the original attributes' distributions using either a Gaussian filter or with a three-bins strategy, where the probability of the true value $T$ is $p(T) \sim \mathcal{U}(p_L, 1), p_L > 0.5$ and the probabilities of its two neighboring values are $p(N_1) \sim \mathcal{U}(0, 1 - p(T))$ and $p(N_2) = 1 - p(T) - p(N_1)$. Note that the motivation behind the three-bins strategy is to introduce variance with minimal additional cost for LRMs' prompt complexity.

## 4. Solving RPMs with LRMs and NeSy probabilistic abductive models

### 4.1. Large reasoning models

We focus our study on the two most prominent SOTA LRMs available to date: the closed-source OpenAI o3-mini model and the open-source DeepSeek R1 model (DeepSeek-AI et al., 2025) (together with its distilled version based on Llama 70B). In Appendix B, we include an additional (limited) comparison between OpenAI o3-mini and its predecessor, OpenAI o1. However, since the performance of the o3-mini model was on par with o1 while costing only a fraction ($\approx 14\times$ less), we decided to experiment only with o3-mini.

We adopt the same evaluation framework used in Hersche et al. (2025) to benchmark LRMs. Contrary to their analysis, however, we focus our investigation on *entangled* prompts (providing all the attributes' values in a single prompt rather than having a single prompt *per* attribute). We are forced to choose this setting because, despite performing worse compared to using *disentangled* prompts (where one separate' prompt is used for each attribute), successive experiments on confounders would have otherwise become trivial. Furthermore, we move from a *predictive approach* (where the model has to generate the missing panel) to a *discriminative approach* (where the model is given a list of candidates and required to choose one among them) (Gendron et al., 2024; Hersche et al., 2025). This choice stems from observing, in the early stages of our evaluation, that LRMs can sometimes pick up valid relations in the input matrices (for instance, relations between the binary encodings of values) which are however not part of the set of rules used in RPM. Unlike the generative approach, the discriminative approach implicitly biases the model into evaluating only the rules defined in RPM without explicitly revealing them, hence reducing the aforementioned issue. For more details on the task, models, and prompting strategy, refer to Appendix A.

### 4.2. NeSy probabilistic abductive reasoning (NeSy-PAR) models

Among the wide spectrum of domain-specific architectures proposed to solve RPM, a growing number of works have recently focused on probabilistic abductive reasoning (Zhang et al., 2021; Hersche et al., 2023; Camposampiero et al., 2024; Sun et al., 2025). Abductive reasoning allows us to selectively infer propositions based on prior knowledge represented in a symbolic form to explain the perceptual observations in the best way (Magnani, 2009).

In this work, we propose an extension to the classical framework of probabilistic abductive reasoning in the form of a novel entropy-based confidence metric to improve its performance when reasoning under uncertainty. In particular, we propose to regularize the contribution to the score/loss of each attribute using the entropy of the confidence values $\mathbf{s}$ (encoding the probability of each rule being the one underlying the behavior of a specific attribute in a RPM panel) used in the abduction step of the framework. Practically, we re-weight the contribution to the loss and score of each candidate panel as

$$\mathcal{L} = \sum_{attr} \frac{\mathcal{L}_{attr}}{H(\mathbf{s}_{attr})} \qquad \mathcal{S} = \sum_{attr} \frac{\mathcal{S}_{attr}}{H(\mathbf{s}_{attr})} \qquad \text{with} \quad H(\mathbf{s}_{attr}) = -\sum_{s \in \mathbf{s}} p(s) \log p(s) \quad (1)$$

where $\mathbf{s} = [s_1, \ldots, s_R]$ is the vector of confidence values in the $R$ rules available to the model (computed from the first two rows of each RPM example) and the attribute losses $\mathcal{L}_{attr}$ and scores $\mathcal{S}_{attr}$ represent the individual attributes' contributions to the training loss

and the candidate prediction metric, respectively. Intuitively, the proposed regularization technique lowers the contribution of attributes whose confidence is uniformly distributed across different rules (which happens when no rule perfectly fits the data and results in high entropy) while increasing the contribution of those attributes for which the confidence in the rule is very concentrated (the model is very confident on a single rule, hence the entropy is low). We implement and evaluate the regularization proposed in Equation (1) within the ARLC model (Camposampiero et al., 2024), one of the SOTA NeSy approaches on RPM.

## 5. Results

### 5.1. LRMs are stronger analogical reasoners than LLMs

Analogical reasoning capabilities of LRMs have not been extensively evaluated to date. In this work, we reduce this knowledge gap by testing this new generation of systems on the well-known benchmark I-RAVEN (Hu et al., 2021), as well as its more difficult extension, I-RAVEN-X (Hersche et al., 2025). Table 1 reports the results for this first proposed evaluation. We additionally include previous results on the closed-source OpenAI GPT-4 (OpenAI et al., 2024) and the open-source Llama-3 70B (Dubey et al., 2024) from Hersche et al. (2025) to allow for a one-to-one comparison between LRMs and LLMs. Firstly, we observe that LRMs can achieve results comparable to LLMs with less engineered prompts and that they generally improve the reasoning accuracy when the level of prompt engineering is on par. o3-mini, for instance, shows no drops in accuracy on I-RAVEN-X and a 6% drop on I-RAVEN compared to GPT-4 while using only $\frac{1}{21}$ of the prompts. When we compare the same two models on similar prompt complexities (that is, using entangled prompting in both settings, but still retaining a $\frac{1}{7}$ ratio between LRMs and LLMs due to self-consistency) o3-mini

| Model | ICL | Prompts | I-RAVEN (3×3) | | | I-RAVEN-X (3×10) | | | | | |
| | | | Range 10 | | | Range 100 | | | Range 1000 | | |
| | | | Task | Arithm. | Tok. | Task | Arithm. | Tok. | Task | Arithm. | Tok. |
|---|---|---|---|---|---|---|---|---|---|---|---|
| Llama-3 70B | ✓ | 21 | 85.0 | 45.0 | 21 | 73.0 | 2.6 | 21 | 74.2 | 0.4 | 21 |
| GPT-4 | ✗ | 21 | 93.2 | 73.6 | 21 | 79.6 | 25.1 | 21 | 76.6 | 8.4 | 21 |
| Llama-3 70B | ✓ | 7 | 79.0 | 31.0 | 21 | 72.6 | 0.0 | 21 | 74.0 | 0.4 | 21 |
| GPT-4 | ✗ | 7 | 74.8 | 27.2 | 21 | 72.8 | 2.7 | 21 | 74.0 | 1.1 | 21 |
| OpenAI o3-mini (medium) | ✗ | 1 | 86.6 | 74.4 | 5445 | 77.6 | 53.2 | 7884 | 81.0 | 60.8 | 7209 |
| OpenAI o3-mini (high) | ✗ | 1 | 92.6 | 86.1 | 9867 | 82.4 | 63.5 | 19041 | 80.6 | 60.1 | 19449 |
| DeepSeek R1 | ✗ | 1 | 80.6 | 74.8 | 4486 | 84.0 | 67.7 | 5550 | 82.8 | 65.8 | 5505 |
| DeepSeek R1 dist. | ✗ | 1 | 78.4 | 69.4 | 5192 | 67.0 | 52.9 | 6690 | 72.0 | 54.4 | 6324 |

Table 1: **Evaluating LRMs on analogical reasoning.** Full task accuracy (i.e, % of test examples correctly predicted) and arithmetic accuracy (i.e., % of the attributes in the test examples governed by an arithmetic relation correctly predicted) of LLMs and LRMs on I-RAVEN and I-RAVEN-X. We report if In-Context Learning (ICL) examples of the task were added to the prompt, the number of total prompts fed into the model (some techniques, such as self-consistency and disentangled prompting require querying the model multiple times), and the number of tokens generated by the model. "Range" indicates the dynamic range of the attributes' values. "Tok." indicates the average number of output tokens of the model. The results for GPT-4 and Llama-3 are taken from Hersche et al. (2025).

emerges as a clear winner, showing a 6.5% increase in accuracy and remarkably stronger performances on arithmetic reasoning. However, this comes at a cost, as shown by the number of output tokens produced by the models during inference, which is two orders of magnitude higher on average compared to LLMs. Secondly, the results show that LRMs are much stronger reasoners than LLMs when challenged with the longer reasoning rules and attribute ranges in I-RAVEN-X. While LLMs show a massive drop in arithmetic accuracy on I-RAVEN-X, nearing 0% for comparable prompt complexity, LRMs are affected by a much smaller arithmetic degradation on average, while sometimes even improving on the overall task accuracy. Overall, we observe that o3-mini and R1 show similar performance on this analogical reasoning task, with o3-mini excelling in the standard I-RAVEN and R1 performing better on I-RAVEN-X. The distilled version of R1, on the other hand, displays weaker results compared to the original model, especially on I-RAVEN-X. To further improve the results on o3-mini, we increased the reasoning effort from `medium` to `high` and set the maximum number of reasoning to its maximum (100,000). The accuracy on I-RAVEN improves by 6%, whereby it stays constant on the most difficult I-RAVEN-X setting despite the increased reasoning effort (2.7× reasoning tokens). Hence, o3-mini (`medium`) was preferred as a cost-efficient solution for the following experiments.

## 5.2. LRMs are significantly challenged by reasoning under uncertainty

The results in Section 5.1 show that LRMs can solve analogical reasoning tasks more accurately than LLMs. However, would they be capable of retaining the same robustness in

| Exp. | Confounders (SNR) | $p_L$ | OpenAI o3-mini | | | DeepSeek R1 | | | ARLC$_{entropy}$ | |
|---|---|---|---|---|---|---|---|---|---|---|
| | | | Task | Arith. | Tok. | Task | Arith. | Tok. | Task | Arith. |
| | 0 ($\infty$) | 1.00 | 81.0 | 60.8 | 7209 | 82.8 | 65.8 | 6324 | 98.3/93.2 | 98.2/97.1 |
| (a) | 1 (4.77) | 1.00 | 76.0 | 53.2 | 11521 | 78.2 | 55.2 | 8919 | 98.5/93.5 | 98.2/97.1 |
| | 3 (0.00) | 1.00 | 75.6 | 51.7 | 11669 | 80.2 | 58.2 | 8429 | 99.0/93.2 | 98.2/97.1 |
| | 5 ($-2.22$) | 1.00 | 71.2 | 48.3 | 12640 | 78.6 | 55.9 | 8681 | 98.6/92.9 | 98.2/97.1 |
| | 10 ($-5.23$) | 1.00 | 69.8 | 45.6 | 13709 | 77.0 | 53.6 | 8912 | 98.8/92.6 | 98.2/97.1 |
| | 300 ($-20.00$) | 1.00 | - | - | - | - | - | - | 97.5/83.1 | - |
| (b) | 0 ($\infty$) | 0.70 | 75.0 | 51.7 | 13112 | 67.4 | 44.9 | 6995 | 92.6/90.4 | 92.2/86.7 |
| | 0 ($\infty$) | 0.51 | 75.6 | 53.2 | 13028 | 63.0 | 46.4 | 7518 | 85.3/83.9 | 84.7/79.5 |
| (c) | 10 ($-5.23$) | 0.51 | 17.0 | 41.1 | 18482 | 23.2 | 45.3 | 7147 | 88.0/82.9 | 88.3/75.4 |
| | 20 ($-8.24$) | 0.51 | - | - | - | - | - | - | 83.2/76.9 | 81.7/62.0 |
| | 30 ($-10$) | 0.51 | - | - | - | - | - | - | 79.2/75.6 | 78.8/61.5 |

Table 2: **Evaluating LRMs and NeSy-PAR models on analogical reasoning under simulated perceptual uncertainty.** Task and arithmetic accuracy (%) on I-RAVEN-X (range [0,1000]) with different numbers of confounders, from 0 (no confounders, SNR=$\infty$) to 10 (SNR=$-5.23$ dB), and different attributes' distribution smoothening (bin-smoothening strategy, with different probabilities assigned to the correct value bin $p_L$). We show experiments with: a) only confounders ; b) only the attributes' distribution smoothing ; c) both confounders and distribution smoothing. For LRMs, we report the number of output tokens to quantify the reasoning effort adopted on average by the model to find a solution. The results for ARLC are reported as max/mean over 5 different random seeds.

scenarios where uncertainty is introduced? To answer this question, we benchmark the two LRMs on the I-RAVEN-X extension proposed in Section 3. We adopt the same methodology used in the previous experiments on I-RAVEN and I-RAVEN-X, with only minor prompting modifications when strictly necessary (e.g., to provide probability distributions for attributes' values). The empirical results of this study are reported in Table 2. Firstly, we observe that LRMs perform significantly worse when noise factors that simulate perceptual uncertainty are integrated into the experiments. For instance, o3-mini's accuracy dropped by 11.2% and 15.2% on task and arithmetic accuracy, respectively, when evaluated with 10 additional confounding attributes. R1, on the other hand, is more robust to confounders (5.8% and 12.2% drops on task and arithmetic accuracy). However, it performs much worse when the attribute values' distributions are smoothened, losing up to 19.8% of task accuracy in the harshest scenario, while o3-mini shows a much smaller degradation (5.4%) in this setting. When both the confounders and distribution smoothening are evaluated together at their maximum level, we observe a sharp drop in task accuracy for both o3-mini (to 17.0%) and DeepSeek R1 (to 22.8%), bringing them close to random chance (12.5%). Moreover, we observe that for o3-mini more challenging perceptual uncertainty conditions directly translate to higher numbers of reasoning tokens (7209 of the base setting to 18,589 of the combined noise experiments). This trend, however, was not observed in R1, where the number of tokens was roughly constant across the different settings. An additional experiment with `high` reasoning effort could only marginally increase the o3-mini's accuracy (to 31.0%) at the cost of 53,596 average reasoning tokens.

### 5.3. NeSy-PAR models are robust when reasoning under uncertainty

We extend the investigation to neuro-symbolic models based on probabilistic abductive reasoning, focusing in particular on ARLC Camposampiero et al. (2024), improved with the entropy regularization introduced in Section 4.2. We report some of these results in Table 2, and more extensive evaluations in Appendix D. ARLC proves to be more robust when reasoning under perceptual uncertainty compared to LRMs, showing no drop in accuracy even in extremely harsh SNR conditions due to confounders thanks to the novel entropy-based regularization (up to $-20\,\mathrm{dB}$ as shown in Appendix D) and maintains its high accuracy when reasoning with smoothened attributes' distributions. Furthermore, when evaluated on the most difficult setting (group (c) in Table 2), ARLC displays much stronger results (88.0% best accuracy compared to 23.2% of the best LRM). Overall, ARLC

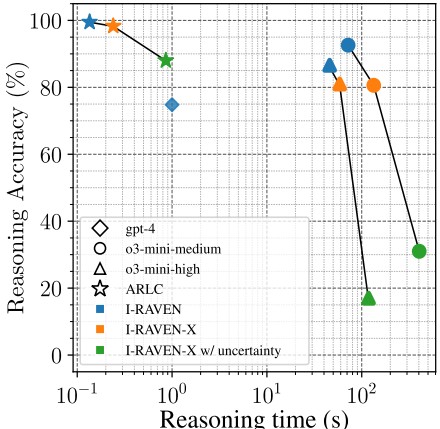

Figure 2: LRM/ARLC reasoning time vs. accuracy.

maintains a remarkably high reasoning accuracy in the trajectory I-RAVEN → I-RAVEN-X, experiencing only a modest decline (98.6% to 88.0%) and significantly outperforming LRMs. ARLC can also successfully learn the set of rules underlying I-RAVEN when trained with highly uncertain attribute distributions, as shown in the ablation included in Appendix D.

Finally, we directly compare ARLC and o3-mini on the accuracy-cost trade-off scaling behavior, as depicted in Figure 2. We use inference time as a proxy for computational cost. The results show that ARLC consistently defines the Pareto frontier, highlighting a more favorable scaling profile compared to o3-mini. See Appendix E for more details.

## 6. Discussion and limitations

Overall, we can draw different conclusions from the experimental results reported in Section 5. First, we confirm that LRMs yield substantial improvements over LLMs also in analogical reasoning tasks, as demonstrated in Section 5.1. Notably, arithmetic reasoning performance shows marked gains, with improvements reaching up to 65.4% in certain settings. However, this favorable trend shifts upon the introduction of simulated perceptual uncertainty. The overall task accuracy drop for LRMs is considerable: o3-mini loses up to 69.6% accuracy and R1 up to 57.2% from the standard I-RAVEN to I-RAVEN-X with uncertainty. Testing on longer reasoning relations and larger attributes' dynamic ranges only plays a smaller part in this (5.6% for o3-mini, and even an increase in accuracy of 2.2% for R1), while perceptual uncertainty accounts for most of the actual drop in accuracy. On the other hand, neuro-symbolic probabilistic abductive reasoning methods show stronger robustness and better scaling laws compared to LRMs. While prior works have circumvented the need for artificially induced perceptual uncertainty by integrating neural perception modules into the reasoning pipeline (Hersche et al., 2023; Shah et al., 2022; Ahrabian et al., 2024a; Jiang et al., 2024a), the unimodal nature of LLMs and LRMs constrains our approach to purely symbolic datasets. On the other hand, we note that the simulated perceptual uncertainty introduced in our setup may be simultaneously overly simplistic and unnecessarily complex. Specifically, the three-bin discretization strategy constitutes an oversimplification of the behavior of a real perception module, which would likely distribute uncertainty across a broader set of elements and introduce additional errors not currently modeled. At the same time, encoding this uncertainty leads to significantly longer and less efficient LRM prompts, which may in turn impair the model's reasoning abilities. We leave a more thorough investigation of these limitations to future work

## 7. Conclusion

This work addresses the lack of support for reasoning under perceptual uncertainty in symbolic analogical reasoning benchmarks. Specifically, it augments an existing benchmark based on RPM, I-RAVEN-X, with confounding attributes and smooth attributes' distributions, that together allow for the simulation of an imperfect perception front-end. This benchmark is then leveraged to evaluate the latest generation of open-domain reasoning systems, Large Reasoning Models (LRMs). Compared to LLMs, LRMs achieve improved productivity for larger reasoning relations and attribute ranges. However, LRMs are still significantly challenged by the (simulated) perceptual uncertainty. On the other hand, neuro-symbolic models based on probabilistic abduction achieve more robust and accurate performance, but cannot directly generalize to different domains in the same way LRMs do. Overall, our results suggest that open-domain, robust analogical reasoning models are still a mirage, and future work has to be invested to achieve this objective.

## Acknowledgments

This work is supported by the Swiss National Science Foundation (SNF), grant 10002666.

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

# Appendices

## Appendix A. Additional details on RPM and prompting

Raven's progressive matrices (RPM) is a visual task that involves perceiving pattern continuation and elemental abstraction as well as deducing relations based on a restricted set of underlying rules in a process that mirrors the attributes of advanced human intelligence (Snow et al., 1984; Snow and Lohman, 1984). In this work, we focus on the I-RAVEN dataset. Each RPM test in I-RAVEN is an analogy problem presented as a $3 \times 3$ pictorial matrix of context panels. Every panel in the matrix is filled with several geometric objects based on a certain rule, except the bottom-right panel, which is left blank. Figure A.3 includes an I-RAVEN example test. The task is to complete the missing panel by picking the correct answer from a set of (eight) candidate answer panels that match the implicit generation rule on every attribute. The object's attributes (color, size, shape, number, position) are governed by individual underlying rules:

- *constant*, the attribute value does not change per row;

- *arithmetic*, the attribute value of the third panel corresponds to either the sum or the difference of the first two panels of the row;

- *progression*, the attribute value monotonically increases or decreases in a row by 1 or 2;

- *distribute three*, the set of the three different values remains constant across rows, but the individual attribute values get shifted to the left or to the right by one position at every row; it also holds column-wise.

Each panel contains a variable number of objects (minimum one, maximum nine) arranged according to one of seven different constellations (center, distribute-four, distribute-nine, left-right, up-down, in-out-center, and in-out-four).

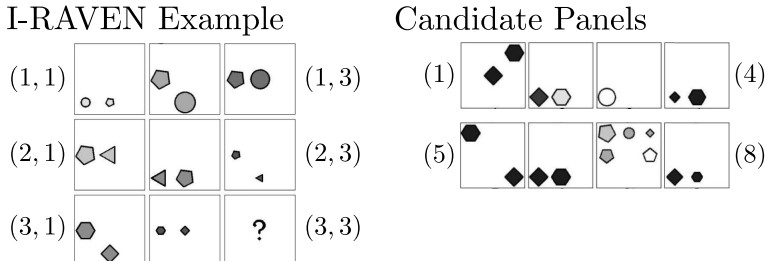

Figure A.3: RPM example from I-RAVEN.

For the OpenAI o3-mini model, we use the `o3-mini-2025-01-31` via the OpenAI API. By default, reasoning efforts were set to `medium` and number of reasoning tokens to 25,000. For DeepSeek R1 model, the full model with 671B parameters was serviced by `www.together.ai`, whereas the distilled version was run locally on 8 NVIDIA A100 GPUs. The maximum number of reasoning tokens was set to 25,000, the temperature to 0.6, and top-p to 0.7. Self-consistency (Wang et al., 2023; Lewkowycz et al., 2022) and attributes' scaling (Hu et al., 2023) were also dropped in the experiments with LRMs. Moreover, no in-context examples of the tasks (Brown et al., 2020) were provided since they were previously observed to be hurtful for LRMs (DeepSeek-AI et al., 2025). We also restrict the investigation to a subset of 500 randomly sampled RPM tests in both I-RAVEN and I-RAVEN-X (due to budget constraints), which we observed to be representative enough of the entire test set (Hu et al., 2023). We report some examples of the prompts used in our experiments in Tables A.3, A.4, A.5, and A.6. The prompting style for embracing CoT was inspired by Wüst et al. (2024). For automatic retrieval of the model's answer, we prompt it to provide its answer in the format "My Answer: Answer #<your answer>". By default, answer panel #0 is predicted if no answer can be retrieved.

---

Complete the Raven's progressive matrix. Your task is to select the correct Answer from the Answer set. Please decide carefully. Take a deep breath and think step-by-step. Finally, give your answer in the following format: My Answer: Answer #<your answer>

row 1: (3,5,5), (6,5,5), (4,5,5);
row 2: (4,3,1), (3,3,1), (6,3,1);
row 3: (6,1,7), (4,1,7),

Answer set:
    Answer #0: (3,2,7)
    Answer #1: (7,1,5)
    Answer #2: (7,2,5)
    Answer #3: (7,2,7)
    Answer #4: (7,1,7)
    Answer #5: (3,1,7)
    Answer #6: (3,2,5)
    Answer #7: (3,1,5)

---

Table A.3: Example prompt for an I-RAVEN task.

Complete the Raven's progressive matrix. Your task is to select the correct Answer from the Answer set. Please decide carefully. Take a deep breath and think step-by-step. Finally, give your answer in the following format: My Answer: Answer #<your answer>

row 1: (6,16,9), (7,15,9), (70,14,9), (93,13,9), (88,12,9), (77,11,9), (83,10,9), (22,9,9), (39,8,9), (27,7,9);

row 2: (7,12,24), (70,11,24), (93,10,24), (88,9,24), (77,8,24), (83,7,24), (22,6,24), (39,5,24), (27,4,24), (6,3,24);

row 3: (70,35,52), (93,34,52), (88,33,52), (77,32,52), (83,31,52), (22,30,52), (39,29,52), (27,28,52), (6,27,52),

Answer set:
    Answer #0: (7,26,52)
    Answer #1: (83,55,52)
    Answer #2: (7,26,37)
    Answer #3: (83,55,37)
    Answer #4: (7,55,52)
    Answer #5: (83,26,37)
    Answer #6: (7,55,37)
    Answer #7: (83,26,52)

Table A.4: Example prompt for an I-RAVEN-X task.

Complete the Raven's progressive matrix. Your task is to select the best matching Answer from the Answer set. Please decide carefully. Take a deep breath and think step-by-step. Finally, give your answer in the following format: My Answer: Answer #<your answer>

row     1:       (917,854,889,837,449,40,616,988,225,603,813,154,860),     (290,853,889,310,920,885,291,416,926,503,379,786,859),
(532,852,889,336,540,95,33,182,41,215,990,859,625),                       (25,851,889,948,465,970,253,795,956,622,323,735,535),
(31,850,889,846,149,643,802,187,413,101,300,378,181),                     (43,849,889,700,975,580,488,662,820,977,189,160,955),
(574,848,889,484,18,951,173,279,247,567,639,939,730),                     (761,847,889,971,245,547,175,991,94,306,976,778,188),
(576,846,889,547,182,955,995,410,545,537,859,368,146), (291,845,889,544,515,965,647,155,660,835,167,363,578);

row     2:       (290,898,875,416,729,621,255,121,775,992,332,824,69),     (532,897,875,617,602,91,626,959,328,566,572,496,129),
(25,896,875,507,14,482,3,638,723,822,326,152,311),                        (31,895,875,551,141,165,894,867,142,856,245,396,325),
(43,894,875,645,712,987,788,382,795,149,295,457,63),                      (574,893,875,269,762,290,698,804,252,56,328,850,702),
(761,892,875,621,590,319,785,4,122,627,517,924,88),                       (576,891,875,268,299,764,678,718,860,626,845,523,1),
(291,890,875,860,69,712,754,590,214,674,171,773,227),  (917,889,875,802,908,433,515,585,256,102,529,939,585);

row     3:       (532,497,831,73,406,82,149,646,932,466,196,966,172),      (25,496,831,76,880,109,467,76,845,392,673,736,51),
(31,495,831,79,825,847,494,174,270,472,649,164,234),                      (43,494,831,39,960,182,917,180,643,977,698,321,467),
(574,493,831,553,583,258,422,840,680,109,870,539,289),                    (761,492,831,481,548,81,43,180,359,410,733,702,708),
(576,491,831,882,329,883,287,624,816,453,120,316,349),                    (291,490,831,398,434,521,426,600,224,181,827,281,512),
(917,489,831,611,791,841,260,28,125,408,122,577,903),

Answer set:
Answer #0: (290,488,875,657,175,669,825,660,980,305,71,297,764)
Answer #1: (851,488,875,785,95,663,714,937,607,543,958,80,215)
Answer #2: (290,451,831,808,72,151,7,665,312,920,665,806,177)
Answer #3: (290,488,831,340,114,819,129,10,922,744,948,540,925)
Answer #4: (851,451,875,714,337,713,987,115,520,218,644,222,463)
Answer #5: (851,488,831,948,251,490,394,977,846,124,951,827,501)
Answer #6: (290,451,875,761,816,59,950,670,732,542,237,552,272)
Answer #7: (851,451,831,9,552,304,979,949,86,118,847,82,575)

Table A.5: Example prompt for the I-RAVEN-X task with confounders.

Complete the Raven's progressive matrix. You are given a context matrix of 3 rows and 10 colums. Each element in the matrix has multiply attributes, embedded in round brackets (). Each attribute is described with a probability distribution, e.g., $<$p_a::v_a, p_b::v_b$>$ describes that the attribute has value v_a with probability p_a and value v_b with probability p_b. Your task is to select the best matching Answer from the Answer set. Please decide carefully. Take a deep breath and think step-by-step. Finally, give your answer in the following format: My Answer: Answer #$<$your answer$>$

row 1: ($<$0.21::916,0.53::917,0.26::918$>$, $<$0.02::853,0.62::854,0.36::855$>$, $<$0.24::888,0.64::889,0.12::890$>$), ($<$0.09::289,0.75::290,0.16::291$>$, $<$0.12::852,0.74::853,0.14::854$>$, $<$0.11::888,0.85::889,0.04::890$>$), ($<$0.44::531,0.55::532,0.01::533$>$, $<$0.36::851,0.63::852,0.01::853$>$, $<$0.24::888,0.74::889,0.02::890$>$), ($<$0.09::24,0.88::25,0.03::26$>$, $<$0.03::850,0.97::851,0.00::852$>$, $<$0.04::888,0.76::889,0.20::890$>$), ($<$0.08::30,0.58::31,0.34::32$>$, $<$0.02::849,0.97::850,0.01::851$>$, $<$-0.00::888,0.91::889,0.09::890$>$), ($<$0.20::42,0.51::43,0.29::44$>$, $<$0.01::848,0.97::849,0.02::850$>$, $<$0.25::888,0.70::889,0.05::890$>$), ($<$0.12::573,0.87::574,0.01::575$>$, $<$0.06::847,0.78::848,0.16::849$>$, $<$0.01::888,0.99::889,0.00::890$>$), ($<$0.04::760,0.82::761,0.14::762$>$, $<$0.08::846,0.70::847,0.22::848$>$, $<$0.04::888,0.77::889,0.19::890$>$), ($<$0.04::575,0.54::576,0.42::577$>$, $<$0.46::845,0.54::846,-0.00::847$>$, $<$0.01::888,0.91::889,0.08::890$>$), ($<$0.15::290,0.85::291,0.00::292$>$, $<$0.04::844,0.78::845,0.18::846$>$, $<$0.30::888,0.66::889,0.04::890$>$);
row 2: ($<$0.01::289,0.81::290,0.18::291$>$, $<$0.19::897,0.59::898,0.22::899$>$, $<$0.20::874,0.72::875,0.08::876$>$), ($<$0.07::531,0.82::532,0.11::533$>$, $<$0.37::896,0.54::897,0.09::898$>$, $<$-0.00::874,0.77::875,0.23::876$>$), ($<$0.12::24,0.72::25,0.16::26$>$, $<$0.01::895,0.78::896,0.21::897$>$, $<$0.34::874,0.66::875,-0.00::876$>$), ($<$0.19::30,0.74::31,0.07::32$>$, $<$0.20::894,0.61::895,0.19::896$>$, $<$0.00::874,0.99::875,0.01::876$>$), ($<$0.20::42,0.77::43,0.03::44$>$, $<$0.02::893,0.95::894,0.03::895$>$, $<$0.08::874,0.73::875,0.19::876$>$), ($<$0.05::573,0.85::574,0.10::575$>$, $<$0.08::892,0.91::893,0.01::894$>$, $<$0.06::874,0.81::875,0.13::876$>$), ($<$0.14::760,0.53::761,0.33::762$>$, $<$0.15::891,0.65::892,0.20::893$>$, $<$0.13::874,0.66::875,0.21::876$>$), ($<$0.05::575,0.65::576,0.30::577$>$, $<$0.01::890,0.82::891,0.17::892$>$, $<$0.12::874,0.66::875,0.22::876$>$), ($<$0.00::290,0.94::291,0.06::292$>$, $<$0.02::889,0.95::890,0.03::891$>$, $<$0.12::874,0.86::875,0.02::876$>$), ($<$0.14::916,0.84::917,0.02::918$>$, $<$0.02::888,0.95::889,0.03::890$>$, $<$0.01::874,0.54::875,0.45::876$>$);
row 3: ($<$0.21::531,0.77::532,0.02::533$>$, $<$0.01::496,0.88::497,0.11::498$>$, $<$0.07::830,0.62::831,0.31::832$>$), ($<$0.20::24,0.79::25,0.01::26$>$, $<$0.19::495,0.62::496,0.19::497$>$, $<$0.06::830,0.92::831,0.02::832$>$), ($<$0.17::30,0.56::31,0.27::32$>$, $<$0.27::494,0.64::495,0.09::496$>$, $<$0.02::830,0.98::831,0.00::832$>$), ($<$0.00::42,0.98::43,0.02::44$>$, $<$0.38::493,0.58::494,0.04::495$>$, $<$0.19::830,0.53::831,0.28::832$>$), ($<$0.07::573,0.52::574,0.41::575$>$, $<$0.01::492,0.99::493,0.00::494$>$, $<$0.01::830,0.81::831,0.18::832$>$), ($<$0.26::760,0.55::761,0.19::762$>$, $<$0.13::491,0.83::492,0.04::493$>$, $<$0.05::830,0.82::831,0.13::832$>$), ($<$0.47::575,0.52::576,0.01::577$>$, $<$0.15::490,0.59::491,0.26::492$>$, $<$0.16::830,0.81::831,0.03::832$>$), ($<$0.03::290,0.82::291,0.15::292$>$, $<$0.29::489,0.52::490,0.19::491$>$, $<$0.03::830,0.85::831,0.12::832$>$), ($<$0.08::916,0.81::917,0.11::918$>$, $<$0.05::488,0.83::489,0.12::490$>$, $<$0.09::830,0.64::831,0.27::832$>$),

Answer set:
Answer #0: ($<$0.06::289,0.83::290,0.11::291$>$, $<$0.00::487,1.00::488,0.00::489$>$, $<$0.03::874,0.82::875,0.15::876$>$)
Answer #1: ($<$0.01::850,0.78::851,0.21::852$>$, $<$0.00::487,0.99::488,0.01::489$>$, $<$0.08::874,0.85::875,0.07::876$>$)
Answer #2: ($<$0.03::289,0.57::290,0.40::291$>$, $<$0.15::450,0.75::451,0.10::452$>$, $<$0.15::830,0.62::831,0.23::832$>$)
Answer #3: ($<$0.06::289,0.52::290,0.42::291$>$, $<$0.03::487,0.92::488,0.05::489$>$, $<$0.31::830,0.61::831,0.08::832$>$)
Answer #4: ($<$0.02::850,0.95::851,0.03::852$>$, $<$0.16::450,0.63::451,0.21::452$>$, $<$0.20::874,0.52::875,0.28::876$>$)
Answer #5: ($<$0.02::850,0.86::851,0.12::852$>$, $<$0.18::487,0.80::488,0.02::489$>$, $<$0.14::830,0.79::831,0.07::832$>$)
Answer #6: ($<$0.01::289,0.96::290,0.03::291$>$, $<$0.38::450,0.59::451,0.03::452$>$, $<$0.08::874,0.68::875,0.24::876$>$)
Answer #7: ($<$0.08::850,0.62::851,0.30::852$>$, $<$0.15::450,0.82::451,0.03::452$>$, $<$0.09::830,0.87::831,0.04::832$>$)

Table A.6: Example prompt for the I-RAVEN-X task with smooth distributions.

## Appendix B. Comparison between OpenAI o3-mini and o1

This Appendix presents a small ablation study on two different closed-source LRMs, OpenAI o1 and OpenAI o3-mini. The goal of these experiments was to measure the difference, if any, in the reasoning capabilities of the o3-mini model compared to its bigger, more expensive predecessor. We restricted the size of the test set to 100 test examples for both I-RAVEN and I-RAVEN-X. The results, presented in Table B.7, show that the two models achieve roughly comparable performance on both I-RAVEN and I-RAVEN-X, with o3-mini being consistently slightly less accurate than o1. However, o1 is also considerably more expensive compared to o3: o1 is priced at \$15 and \$60 per million input and output tokens, respectively, while o3-mini costs only \$1.1 and \$4.4 per million input and output tokens (approximately 14× less expensive). Hence, we opt to use only o3-mini in the full evaluation.

| Model | Setting | I-RAVEN | | I-RAVEN-X | | | |
| | | Range 10 | | Range 100 | | Range 1000 | |
| | | Task | Arithm. | Task | Arithm. | Task | Arithm. |
| --- | --- | --- | --- | --- | --- | --- | --- |
| OpenAI o1 | Entangled | 88.0 | 79.7 | 86.0 | 68.2 | 86.0 | 68.2 |
| OpenAI o3-mini | Entangled | 86.6 | 81.4 | 84.0 | 63.6 | 81.0 | 60.8 |

Table B.7: Task and arithmetic accuracy (%) comparison of two different LRMs on a subset of 100 test examples of I-RAVEN and I-RAVEN-X.

## Appendix C. I-RAVEN noisy attributes are not noisy

This Appendix highlights one major limitation of the so-called *noise* attributes in RAVEN and I-RAVEN (Orientation and Uniformity). These attributes, in reality, do not introduce any noise in the reasoning process for two reasons:

- these attributes' values always respect one of the underlying rules of RAVEN (e.g., in the example shown in Figure C.4, Orientation can be inferred using the constant rule); hence, they do not introduce any noise if used along the other main attributes to learn the rules of RAVEN in a data driven fashion;

- at inference time, these attributes do not reduce the signal-to-noise ratio of of the RAVEN examples and do not change the probability distribution over the candidate panels (e.g., in Figure C.4 all candidates are equally likely, and this will not influence the final prediction of the answer panel).

As a result, these attributes alone do not increase the difficulty of RAVEN on their own. The confounders introduced for I-RAVEN-X in Section 3 address both problems, being sampled at random in the dynamic range of each attribute.

$$
\begin{array}{|ccc|}
\hline
2 & 2 & 2 \\
4 & 4 & 4 \\
7 & 7 & ? \\
\hline
\end{array}
\qquad
\begin{array}{|ccccccc|}
\hline
7 & 7 & 7 & 7 & 7 & 7 & 7 \\
\hline
\end{array}
$$

Figure C.4: Example of the Orientation attribute in I-RAVEN, showing an example $3 \times 3$ on the left and the eight candidate panels on the right.

## Appendix D. Additional experimental results for ARLC

This appendix presents additional results on our neuro-symbolic baseline, ARLC, which were not included in the main manuscript for space constraints.

Firstly, we ablate the effectiveness of the entropy regularization proposed in Section 4.2 by integrating it in ARLC and comparing this improved version with the vanilla counterpart of the model. We perform this ablation on two different settings:

1. **training and inference with confounders**, to test whether the model can still learn the correct set of rules underlying RAVEN examples in settings with noisy supervision;

2. **training on clean data, inference with confounders**, to test the inference-time robustness of the model.

Naturally, the setting where the model is also trained on confounding attributes is more challenging, as the training signal that can be used to learn the rules underlying the task linearly decreases in the number of confounding attributes used. We report the results of this ablation in Table D.8 on both dynamic ranges supported by I-RAVEN-X. As it can be observed from the results, entropy regularization is significantly helpful both when used as training+inference or inference-only technique. In the latter case, it becomes increasingly more effective compared to the vanilla model as the number of attributes increases.

To stress the robustness of the proposed entropy regularization, we also test under extreme noise conditions ($-20$ dB, 300 confounder attributes). Since the evaluation of the model with this many attributes starts becoming increasingly expensive, we limit the evaluation to the 1000 range subset and reduce the number of different seeds used from 5 to 3. We observe that, while the average task accuracy starts dropping, some of the runs can

| Training data | Entropy | Confounders | SNR | I-RAVEN-X | | | |
| --- | --- | --- | --- | --- | --- | --- | --- |
| | | | | Range 100 | $\Delta$ | Range 1000 | $\Delta$ |
| Noisy | ✗ | 0 | $\infty$ | 100.0/96.9 | - | 98.8/94.0 | - |
| | ✗
✓ | 5 | $-2.2$ | 90.6/83.1
99.3/88.3 | -
+8.7/5.2 | 88.2/80.1
99.1/85.6 | -
+10.9/5.5 |
| | ✗
✓ | 10 | $-5.2$ | 85.7/78.4
94.6/88.3 | -
+8.9/9.9 | 83.6/76.9
92.1/86.0 | -
+8.5/9.1 |
| Clean | ✗ | 0 | $\infty$ | 100.0/96.9 | - | 98.8/94.0 | - |
| | ✗
✓ | 5 | $-2.2$ | 95.3/94.2
100.0/96.4 | -
+4.7/2.2 | 95.3/91.7
98.6/92.9 | -
+3.3/1.2 |
| | ✗
✓ | 10 | $-5.2$ | 93.7/91.4
100.0/96.0 | -
+6.3/4.6 | 92.5/89.5
98.8/92.6 | -
+6.3/3.1 |
| | ✗
✓ | 30 | $-10$ | 90.5/82.0
99.5/94.3 | -
+9.0/12.3 | 88.4/82.0
98.7/92.0 | -
+10.3/10 |
| | ✓ | 300 | $-20$ | - | - | 97.5/83.1 | - |

Table D.8: Ablation of the proposed entropy regularization method using neuro-symbolic ARLC.

still achieve remarkable accuracy, indicating that this technique potentially enables probabilistic abductive reasoning models to work well in settings where only a tiny fraction of the extracted attributes are important for the reasoning task.

On top of the experiments on smoothened distributions included in the main text, we also study the robustness of ARLC when the input distributions are perturbed using a Gaussian filter. This represents a more general setting compared to the three-bin smoothening strategy adopted in the main text, which was primarily chosen to limit the complexity of the prompt for LRMs. In particular, we smoothen the input distribution using the Gaussian filter

$$G(x) = \frac{1}{\sqrt{2\pi\sigma^2}} \exp\left(-\frac{(x-\mu)^2}{2\sigma^2}\right)$$

where $\mu$ corresponds to the index of the true value and $\sigma$ is a tunable parameter that regulates how flat the resulting distribution is. We study different settings and combinations of training and inference perturbations to gain a more comprehensive picture of the behavior of ARLC in this setup. In particular, we evaluate three separate settings:

1. **training and testing with noisy distributions**, to understand how the model would behave in settings where uncertainty on the attributes' values is always present;

2. **training on noisy distributions and evaluating on clean data**, to evaluate whether the model can learn valid rules from noisy data;

3. **training on clean data and evaluated with noisy distributions**, to understand how well a model that learned the correct rules underlying RAVEN would perform when evaluated with an imperfect perception front-end.

We report the results of this ablation in Table D.9. Different interesting observations can be made on these data. Firstly, we observe that training on noisy data and evaluating on clean data always results in competitive performance. This is a clear indication that, despite

| $\sigma$ | Training | Inference | I-RAVEN $(3 \times 3)$ | I-RAVEN-X $(3 \times 10)$ | |
|---|---|---|---|---|---|
| | | | Range 10 | Range 100 | Range 1000 |
| 0.0 | Clean | Clean | 99.8/98.4 | 100.0/96.9 | 98.8/94.0 |
| 0.3 | Noisy | Clean | 99.3/98.1 | 99.8/91.3 | 99.4/89.7 |
| | Clean | Noisy | 96.4/92.2 | 91.6/85.0 | 92.1/89.0 |
| | Noisy | Noisy | 94.6/92.2 | 89.2/76.7 | 97.3/86.4 |
| 0.5 | Noisy | Clean | 99.0/98.4 | 98.9/88.6 | 98.7/86.0 |
| | Clean | Noisy | 86.8/80.3 | 79.2/77.8 | 85.8/83.1 |
| | Noisy | Noisy | 86.0/81.1 | 87.2/74.7 | 90.0/78.7 |
| 0.7 | Noisy | Clean | 98.9/93.0 | 99.0/87.5 | 98.3/85.1 |
| | Clean | Noisy | 64.9/58.8 | 71.9/70.0 | 79.9/77.4 |
| | Noisy | Noisy | 67.6/58.0 | 69.8/64.7 | 84.6/74.1 |

Table D.9: Gaussian smoothening of the input distributions.

the noise in the training process, ARLC can still learn a valid set of rules that yield good results on de-noise data. Even if the mean accuracy shows a degradation proportional to $\sigma$ (expected, since in the harshest settings the probability of the true value is lower than the sum of all the other probabilities of the non-true values), we can still recover close to perfect accuracy in some runs, as shown by the max accuracy reported for these experiments. This is encouraging since model selection using a validation split would allow us to identify and select the models that learned the best set of rules during training.

On the other hand, evaluating models trained on clean data (that learned a good set of rules) with smoothed attributes' distributions sensibly degrades the test accuracy, especially for larger values of $\sigma$. Unfortunately, not much can be done to address this. However, this is still an interesting result, as it underlines the importance of a confident front-end perception in analogical reasoning, showing that excessively flattened attributes' distribution can seriously undermine the accuracy of the reasoning process. Finally, we observe that training with smoothened distributions generally yields better results (at least in the maximum test accuracy) than training on clean data and evaluating with smoothened distributions. This might suggest that, sometimes, integrating uncertainty in the training process can increase the robustness of the model at inference time and guarantee better performances compared to models that have trained exclusively on clean data.

## Appendix E. Accuracy-cost scaling behavior experiments

In this section, we include additional details on Figure 2, reporting the accuracy-cost tradeoff for different models: o3-mini with different reasoning budgets (medium and high), GTP-4o, and ARLC. We opt to quantify cost in terms of average inference time to solve a test example; some of the models (e.g., OpenAI o3-mini) are, in fact, only accessible through API, and an exact quantification of the computation resources required by the inference is not publicly available. For ARLC, the time measurements are performed accelerating the model on a NVIDIA A100-40G GPU using batch size 1.

