# OpenReview forum: "Can Large Reasoning Models do Analogical Reasoning under Perceptual Uncertainty?"
_nesyconf.org/NeSy/2025/Conference — NeSy 2025 Poster_

### Official Review · Reviewer_LSfR · 2025-04-04
**No, they are not solving visual analogical reasoning.**

**Rating:** 6
**Confidence:** 3

**Review:**

This work presents an evaluation of two state-of-the-art Large Reasoning Models (LRMs), OpenAI’s o3-mini and DeepSeek R1, on analogical reasoning, focusing on well-established nonverbal human IQ tests based on Raven’s progressive matrices.
A neuro-symbolic probabilistic abductive model, ARLC, is tested when reasoning under I- RAVEN, scoring 88.0%.

Weaknesses:

I disagree with some of the framing: the input representation in I-RAVEN and I-RAVEN-X is not a visual reasoning query. It has been translated to some numerical concepts that LRM can interpret as patterns but there is no justification that this translation align with human interpretation of a RAVEN's test and that represent visual analogical reasoning.
The "perceptual uncertainty" included in I-RAVEN-X seems an ad-hoc function which works for the experimentation presented.

**Anonymity:**

Remain anonymous

---

### Official Review · Reviewer_UFrd · 2025-04-06

**Rating:** 6
**Confidence:** 5

**Review:**

This paper evaluates Large Reasoning Models (LRMs): OpenAI o3-mini and DeepSeek R1, on Raven's progressive matrices task. The authors extend a recently introduced dataset, I-RAVEN-X, with two approximations of perceptual uncertainty. Then, they also extend a neuro-symbolic probabilistic model called ARLC with entropy-based losses.

The evaluation of LRMs compared to LLMs on I-RAVEN-X, and especially its extension, is relevant and informative. Moreover, it is relevant for the NeSy community that the extended ARCL model is able to perform more robustly on this task. The ablation of this method in the Appendix provides further support for its consistent positive effect. Besides the value of its empirical insights, the paper is fairly well written, and most of its content is relatively easy to follow. I also understand that the space limitations are a cause for some of the brevity that hurts the readability of the paper, hence my judgment is slightly leaning to the positive scale.

Weaknesses and comments:
1. Novelty - The contributions of this work are not explicitly enumerated. Claims of novelty are stated implicitly, based on which I infer that the contributions are: a) first evaluation of LRMs on an analogical reasoning task; b) an extension of a benchmark to consider perceptual uncertainty, unlike in prior work; and c) an extension of ARLC to fit the task in question.
a) Given that the LRMs in question are relatively recent, it is likely that this work is one of the first to evaluate their analogical abilities. I only wonder where whether the "extensively evaluated" claim is not overselling the contribution.
b) Regarding b, to my knowledge, the authors are right that the controlled introduction of uncertainty is novel. However, note that a line of other works, such as [Jiang et al., 2024] and [Ahrabian et al., 2024], expect the models to extract relevant perceptual signal on their own, which naturally yields uncertainty. I was surprised that these studies were not compared against in §2, I would be curious to see such comparison. Moreover, it is not apparent why it is more beneficial to test introducing arbitrary noise and confounders rather than using existing perception models that would naturally introduce such uncertainties.
c) The adaptation of the method ARLC is difficult to follow, since the description of the original method is not self-contained.

2. Clarity - The paper is sometimes hard to follow due to complex terminology that is unclear or insufficiently supported. E.g., the introduction talks about combining "all its productivity and robustness measures", about "smoothened value distributions"; §2 talks about "Neuro[Symbolic] systems (type 6)"; §4.1 states that the 500 samples are "representative enough" though it is unclear based on what; §5 talks about results without introducing the metrics: these must be then deduced in passing from the table captions or eventually from the discussion, and even then the notions of "task" vs. "arithmetic" accuracy are not defined; in §5, it is not clear to me what is meant by "the dynamic range of the attributes' values".

3. Generalization - It would be good if the authors discuss briefly the strengths and limitations of the ARLC-adapted method. Namely, given that its performance is nearly perfect in most scenarios, should we assume this is the go-to method for abstract visual reasoning puzzles? Would the method generalize to new puzzle configurations, attributes, etc.?

**Anonymity:**

Remain anonymous

---

### Official Review · Reviewer_krne · 2025-04-07
**Incremental Additions to the Interesection of NeSy and RPMs**

**Rating:** 7
**Confidence:** 4

**Review:**

The submission presents experimental results for analogical reasoning benchmarks for the modern "Large Reasoning Models" OpenAI o3 mini and DeepSeek R1. These benchmarks include simulated perceptual uncertainty. The analysis also includes a Neuro-Symbolic baseline, ARLC, extended with an entropy-based confidence metric to reflect this uncertainty. The authors show that under uncertainty, LRMs see a much larger dip in discriminative accuracy compared to the ARLC baseline.

 ### Quality

 The submission shows mostly good scientific practice.

 I am unsure about the choice of using I-RAVEN-X over I-RAVEN apart from the fact that the I-RAVEN-X dataset has been presented by the same authors. On the one hand, this new dataset reduces the comparability to the many existing results on the RAVEN-family of datasets, as only a limited amount of approaches have been benchmarked on it. Limiting the matrices to the "center" layout also removes the need for flexibility in the architecture (although I do acknowledge that all architectures presented here have also been tested on I-RAVEN), which allows architectures to take certain shortcuts (e.g. Shah et al. or Zhao et al., as cited below). Thirdly, increasing the matrix size may have implications which haven't been tested yet. Proper experiments on the behavior of I-RAVEN-X e.g. with humans and the effect of matrix size should be tested before introducing additional confounding variables (I mention this here because the work is done by the same authors).

The references to Neuro-Symbolic approaches for RPM problems has significant gaps, as it focuses only on the abductive reasoning approaches pursued by the authors. For example, publications such as Zhao et al. (2023) or Shah et al. (2022) take an appraoch starting from the visual inputs rather than symbolic interpretations, bypassing the need for artificial uncertainty.

- Zhao, S., You, H., Zhang, R. Y., Si, B., Zhen, Z., Wan, X., & Wang, D. H. (2023). An interpretable neuro-symbolic model for raven’s progressive matrices reasoning. _Cognitive Computation_, _15_(5), 1703-1724.

- Shah, V., Sharma, A., Shroff, G., Vig, L., Dash, T., & Srinivasan, A. (2022). Knowledge-based analogical reasoning in neuro-symbolic latent spaces. _arXiv preprint arXiv:2209.08750_.

It should be noted, however, that both of these architectures are, to a degree, modular, and that the presented dataset extension may help in focusing developing of the reasoning engine of such architectures and abstracting from the perceptual component.

Overall, having worked in feature extraction from visually presented RPMs myself, the proposed addition of uncertainty is still an oversimplification of the practical difficulty of this task when trying to build a general model and keep assumptions minimal.

I would question whether the entirety of the accuracy dip is due to the LRM's reasoning capabilities, or whether some of it is due to inefficient prompts, as the ones presented in the Appendix do already seem convoluted, to a point where I can imagine combining the number input of both may in fact impact model performance. The prompt choice is only shortly discussed on page 6, and only the fact that disentangling the prompt would trivialize the introduction of noise attributes.

Introducing uncertainty among only three options seems like a grave oversimplification. It is even explained on page 6 that the reason behind it is that the prompts would get too complex (maybe indicating inefficient prompts? – or perhaps even justifying the otherwise weaker multimodal models).

While it is claimed that the multimodal models perform worse than the textual ones, there has been no comparison drawn from the uncertain textual data to simple visual input (which would make sense here, i.e. "is it worth to build a perceptual module, or should you just use the multimodal model"). I realize that I-RAVEN-X does not have a visual component, but that would just be another reason to stick with I-RAVEN.

Doing so many _different_ things in a short paper (more on that in section clarity) should never come at the cost of a reflected discussion of the presented approaches and results, which is completely missing from this submission. The above mentioned limitations are, in my opinion, fairly obvious and could've already been addressed in the original submission.

### Clarity

The presented submission is well written and reasonably structured. Figures and tables are easy to understand, well captioned, and sensibly referenced.

The submission, however, opens too many discussion points to conclude in such a short paper, resulting in left out details and lack of a clear narrative, which makes my opening summary as convoluted as it is.

In addition, the submission is difficult to interpret without being fully in the loop on the author's previous publications. I am very confident in this statement as I am very familiar with the literature on neuro-symbolic approaches for Abstract Visual Reasoning, specifically RPMs, and still found it difficult to interpret the implications of some of the results presented here.

Importantly, on page 5, Kingma and Welling's paper "Auto-Encoding Variational Bayes" is cited wrongly as published in 2022, which was merely the most recent update to the ArXiV entry. It **must** be corrected to 2013, which was the original submission year, and locates the referenced work correctly in the research timeline.

### Originality

The submission presents two rather incremental additions with the artificial noise to an existing dataset and the inclusion of entropy in the loss function for a _specific_ architecture. Paired with the experiment on LRMs, the paper prevents some – if limited – novelty.

### Significance

While the two presented approaches themselves are only small steps, they may potentially have a large impact on research in the fields of Neural Symbolic Artificial Intelligence and Abstract Visual Reasoning.

### Pro

- The submission shows mostly good scientific practice.
- From an experimental perspective, simulating uncertainty can be very helpful for developing reasoning systems without having to worry about the visual part of the AVR process.
- The presented approach for considering uncertainty in ARLC may be used to generally account for uncertainty over discrete inputs for architectures working on similar inputs.

### Con

- The submission is missing a reflected discussion.
- The submission opens too many discussion points for its length and does not bring them to a conclusion in enough detail.
- The presented approach to perceptual uncertainty does not fully capture the complexity of extracting relevant attributes from visual reasoning premises.

**Anonymity:**

Remain anonymous